# A wheeze recognition algorithm for practical implementation in children

**Chizu Habukawa**[1]*, **Naoto Ohgami**[2], **Naoki Matsumoto**[3], **Kenji Hashino**[3], **Kei Asai**[2], **Tetsuya Sato**[2], **Katsumi Murakami**[4]

1 Department of Paediatrics, Minami Wakayama Medical Center, Wakayama, Japan, 2 Clinical Development Department, Technology Development HQ, Development center, Omron Healthcare Co., Ltd, Kyoto, Japan, 3 Core Technology Department, Technology Development HQ, Development Center, Omron Healthcare Co., Ltd, Kyoto, Japan, 4 Department of Psychosomatic Medicine, Sakai Sakibana Hospital, Osaka, Japan

* gd6c-hbkw@asahi-net.or.jp

**Data Availability Statement:** Data cannot be shared publicly because of this study is human subject research. Data are available from the Omron Healthcare Co., Ltd. for researchers who meet the criteria for access to confidential data.

## Abstract

### Background

The detection of wheezes as an exacerbation sign is important in certain respiratory diseases. However, few highly accurate clinical methods are available for automatic detection of wheezes in children. This study aimed to develop a wheeze detection algorithm for practical implementation in children.

### Methods

A wheeze recognition algorithm was developed based on wheezes features following the Computerized Respiratory Sound Analysis guidelines. Wheezes can be detected by auscultation with a stethoscope and using an automatic computerized lung sound analysis. Lung sounds were recorded for 30 s in 214 children aged 2 months to 12 years and 11 months in a pediatric consultation room. Files containing recorded lung sounds were assessed by two specialist physicians and divided into two groups: 65 were designated as "wheeze" files, and 149 were designated as "no-wheeze" files. All lung sound judgments were agreed between two specialist physicians. We compared wheeze recognition between the specialist physicians and using the wheeze recognition algorithm and calculated the sensitivity, specificity, positive predictive value, and negative predictive value for all recorded sound files to evaluate the influence of age on the wheeze detection sensitivity.

### Results

The detection of wheezes was not influenced by age. In all files, wheezes were differentiated from noise using the wheeze recognition algorithm. The sensitivity, specificity, positive predictive value, and negative predictive value of the wheeze recognition algorithm were 100%, 95.7%, 90.3%, and 100%, respectively.

The dataset in this paper contains the breathing sounds of the patient and ambient sounds, which include the voices of medical staff and patients, corresponding to personal information. We have obtained the consent of all participants as below: "The personal information are not disclosed when the research results are published". Therefore, we cannot disclose it. This was reviewed by the Minami Wakayama Medical Center's Ethics Review Committee [approval number 2016-22(2) approval date January 19, 2016]. If you have request, you can contact to Naoko Fukuda who is contact personnel of the Minami Wakayama Medical Center's Ethics Review Committee (fukuda.naoko. ua@mail.hosp.go.jp).

**Funding:** The Omron Health Care Corporation provided support in the form of salaries for authors Naoto Ohgami, Naoki Matsumoto, Kenji Hashino, Kei Asai and Tetsuya Sato, but did not have any additional role in the study design, data collection and analysis, decision to publish, or preparation of the manuscript. The specific roles of these authors are articulated in the "author contributions" section. This does not alter our adherence to the policies of PlosOne on sharing data and materials. Chizu Habukawa and Katsumi Murakami received a research grant from Omron Health Care Corporation.

**Competing interests:** Chizu Habukawa, Medical Doctor in Minami Wakayama Medical Center: Financial support (Omron Healthcare), Patent. Naoto Ohgami, Employee of Omron Healthcare Co., Ltd.: Employee (Omron Healthcare), Patent. Naoki Matsumoto, Employee of Omron Healthcare Co., Ltd.: Employee (Omron Healthcare), Patent. Kenji Hashino, Employee of Omron Healthcare Co., Ltd.: Employee (Omron Healthcare), Patent. Kei Asai, Employee of Omron Healthcare Co., Ltd.: Employee (Omron Healthcare), Patent. Tetsuya Sato, Employee of Omron Healthcare Co., Ltd.: Employee (Omron Healthcare). Katsumi Murakami, Medical Doctor in Sakai Sakibana Hospital: Financial support (Omron Healthcare), Patent. This does not alter our adherence to the policies of PlosOne on sharing data and materials.

## Conclusions

The wheeze recognition algorithm could identify wheezes in sound files and therefore may be useful in the practical implementation of respiratory illness management at home using properly developed devices.

## Introduction

Wheezes are the most important exacerbation sign in various respiratory diseases among all age groups [1–3] and are usually observed in children with asthma but also noted in those with bronchiolitis and other diseases [4–6]. In most cases, wheezes continuously occur over a prolonged period and should, therefore, be treated and managed before the patient's respiratory condition deteriorates [7, 8].

Therefore, even mild wheezes should properly be detected to identify appropriate treatment. Wheezes are recognized as a significant auscultatory sound during the assessment of respiratory conditions using a stethoscope [1–3]. Results of auscultation are highly subjective and cannot be easily reproduced because of differences in doctors' experience and perceptual abilities, which cause differences in assessment [4, 7, 9–11].

Adult patients with asthma usually self-test at home using a peak flow meter to detect mild attacks; however, children cannot perform sufficiently accurate self-testing with this device [12]. Physicians are almost entirely dependent on reports from family members and caregivers, who may evaluate wheeze differently to a physician [9–11]. Therefore, an objective method to detect wheezes would be beneficial for physicians and patients' families or caregivers.

Computerized lung sound analysis, especially computerized wheeze detection, is a more objective and standardized method, which can overcome the limitations of subjective auscultation [3, 6]. In 1995, Gavriely published the details of a technological approach for automated digital data acquisition and processing of breathing sounds. The commercial device that he developed, PulmoTrack®, enabled automated and continuous monitoring of wheezes [13]. Prodhan et al. used this monitor in a pediatric intensive care unit and reported that wheezes detection by PulmoTrack® was more accurate compared with detection by hospital staff [14]. Boner et al. reported that monitoring wheezes during sleep was useful when treating children with asthma and that TwTtot (the duration of wheezes during the recording) correlated with peak expiratory flow rate changes [15]. Therefore, automated wheeze detection may be useful for the management of children with wheezes, especially infants.

Few reported algorithms were available for clinical use in noisy environments, such as a pediatric consultation room or at home, which can automatically recognize wheezes [16]. In addition, the examination time should be <30 s, because small children cannot endure long examination times without crying or moving. Hence, we aimed to develop an algorithm for automatic wheeze recognition based on its characteristics for clinical use in young children, including infants.

## Materials and methods

### Participants

All participants were outpatient children who attended at the Minami Wakayama Medical Center, the secondary emergency general hospital of the National Hospital Organization located in Wakayama, Japan, between September 27, 2016, and October 25, 2017. All

participants were brought into the hospital for the treatment of recurrent wheezes with cough and dyspnea. Written informed consent was obtained from all participants or their legal guardians. The study protocol was approved by the ethics committee of the hospital [approval number 2016-22(2); approval date: January 19, 2016].

All participants aged ≥ 6 years were diagnosed with bronchial asthma, and their asthmatic severities were classified as mild asthma according to the Japanese Pediatric Guideline for the Treatment and Management of Asthma 2017 [17]. These participants were treated with leukotriene receptor antagonist and/or inhaled corticosteroid in accordance with the guidelines [17]. Participants aged ≤ 5 years had reported at least two episodes of wheezes and had been treated with leukotriene receptor antagonist or without medicine for long-term management. Other chronic diseases, including chronic sinusitis, laryngomalacia, whooping cough, immunodeficiency, cardiac or neonatal pulmonary problems, infantile lung diseases, and gastro-esophageal reflux disease, were excluded by a physician specializing in childhood respiratory and allergic diseases.

### Study procedures

A specialist physician examined all participants using a stethoscope and simultaneously recorded lung sounds during tidal breathing in the pediatric consultation room for at least 30 s. Recordings were obtained from the upper right anterior chest region at the second intercostal space in the midclavicular line of the chest wall. The recorded lung sounds (with or without wheezes) were then listened to independently by two specialist physicians and then confirmed and classified in accordance with the previous methods. Judgments were agreed between two physicians. If the judgments of the two physicians did not match, the supervisor made the judgment [18].

A total of 65 participants' recordings were designated as "wheeze" files, and 149 were designated as "no-wheeze" files. In addition, two specialist physicians differentiated wheezes from lung sound samples, including inspiratory and expiratory lung sounds, nasal congestion, crying, and voices.

### Sound recording and analysis

The sound recording system consisted of a handheld assembled microphone unit (Omron Healthcare Corporation, Kyoto, Japan) and a pulse code modulation recorder (Sony Corporation, Tokyo, Japan). The microphone unit comprised two microphones (ST Microelectronics, Geneva, Switzerland): one recorded ambient sounds around the device (environment microphone), and the other recorded lung sounds from the right side of the chest (lung sound microphone).

Sounds picked up by the microphones (digital signals) were processed (filtered and amplified) by a central processing unit and converted to analog signals by a digital-to-analog converter. This signal was then recorded using the pulse code modulation recorder.

Following the Computerized Respiratory Sound Analysis guidelines, the dominant frequency of a wheeze is > 100 Hz and the duration is > 100 ms [19]. Furthermore, a previous report described the frequency range of a typical wheeze as between 100 and 5000 Hz [20]. The maximum duration of a wheeze is within the expiratory duration. A wheeze detection algorithm was developed based on this definition.

Fig 1 (right panel) shows a typical wheeze spectrogram, with time (s) and frequency (Hz) on the horizontal and vertical axes, respectively. The sound intensity (dB) is shown as color and brightness. A continuous wheeze spectrum was created on the basis of the lung sound analysis. On the left panel, the horizontal axes show intensity (dB) and the vertical axis shows

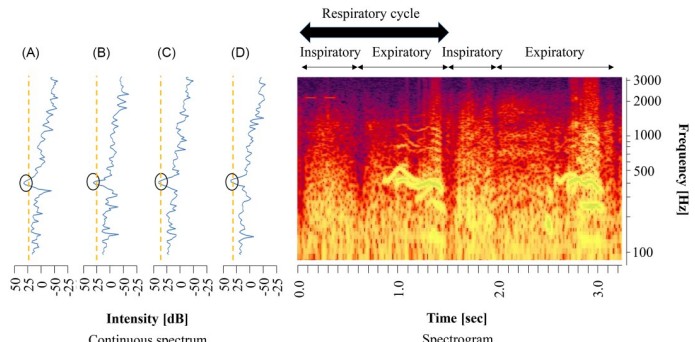

**Fig 1. Spectra and spectrograms of wheezes in lung sounds.** (a) Spectrum 1: Fast Fourier transform (FFT) frame, 0.682–1.054 s; (b) Spectrum 2: FFT frame, 0.700–1.072 s; (c) Spectrum 3: FFT frame, 0.718–1.090 s; (d) Spectrum 4: FFT frame, 0.736–1.108 s.

frequency (Hz). Wheeze sounds are shown as horizontal bars with intensity corresponding to peaks in the power spectrum display [19–22].

## Wheeze recognition algorithm

Based on the wheezes characteristics shown in Fig 1 (right panel), a flowchart was created for the developed wheeze recognition algorithm from sound collection to result generation (Fig 2).

The overall approach consisted of five phases. Details of the processing scheme are showed as follows:

Step 1, sound data were preprocessed using high and low bandpass filters. Data were resampled at a sampling rate of 11.025 kHz and a 16-bit quantization rate.

Step 2, Fast Fourier transform (FFT), the most popular acoustic analysis method, was used. FFT analyzes the intensity for each frequency of sound data. The sound data were preprocessed in a Hamming window of 4096 points (372 ms), and processing was repeated each time the sound data increased by 128 points (18 ms) [23–25].

Step 3, as shown in Fig 1 (left panel), the lung sound spectra had many local maximum points in each time. Therefore, some local maximum points higher than the threshold were extracted as candidates for wheezes sounds. Black-circled points indicate the extracted local maximum points. The orange dotted line represents the threshold value used to determine the

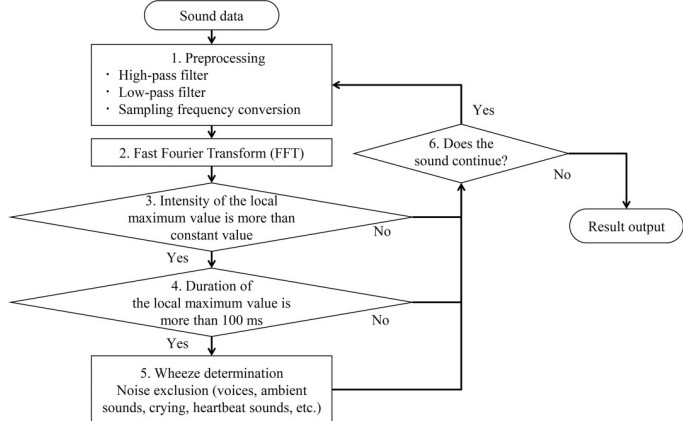

**Fig 2. Flowchart of the newly developed wheeze recognition algorithm for children.**

local maximum point. Threshold values were determined from overall sound pressure levels between 90 and 5000 Hz.

Step 4, whether the local maximum points selected in step 3 continued for $> 100$ ms was determined according the definition of wheeze characteristics [19]. Continuous local maximum values selected in step 4 still included wheeze sounds and other noises: voices, ambient sounds, crying, and heartbeat sounds.

Step 5, threshold values were determined from feature values in order to eliminate noises. To finally determine the presence of wheezes using both lung and ambient sounds, the feature values of the wheezing candidates selected in step 4 were calculated, and the classification model was created using a machine-learning method (decision tree). Finally, if at least one wheeze sound was heard in a file, the file was identified as a wheeze file, whereas if no wheeze sound was heard in a file, it was identified as a no-wheeze file.

For validation, we compared the results of wheeze sound recognition using the algorithm to the results of all files that were discriminated by two specialist physicians.

## Statistical analysis

The results fell into one of the following four categories: actual positives that were correctly predicted as positives (true positives, TP); actual positives that were wrongly predicted as negatives (false negatives, FN); actual negatives that were correctly predicted as negatives (true negatives, TN); and actual negatives that were wrongly predicted as positives (false positives, FP). We analyzed the sensitivity, specificity, positive predictive value (PPV), and negative predictive value (NPV) using the results of the wheeze recognition algorithm in all data files [26, 27]. PPV is the probability that files that were identified as "wheeze" files by the specialists were also identified as "wheeze" files by the algorithm. PNV is the probability that files that were identified as "non-wheeze" files by the specialists were also identified as "non-wheeze" files by the algorithm.

$$Sensitivity = \frac{number\ of\ True\ Positive}{number\ of\ Condition\ Positive} = \frac{TP}{TP + FN}$$

$$Specificity = \frac{number\ of\ True\ Negative}{number\ of\ Condition\ Negative} = \frac{TN}{TN + FP}$$

$$PPV = \frac{number\ of\ True\ Positive}{number\ of\ Predicted\ condition\ Positive} = \frac{TP}{TP + FP}$$

$$NPV = \frac{number\ of\ True\ Negative}{number\ of\ Predicted\ condition\ Negative} = \frac{TN}{TN + FN}$$

Statistical analysis was performed using R software, version 3.4.1 [28]. The patient characteristics were presented as the mean ± range. Characteristics of wheeze sounds were presented as mean ± SD and range. The ratios of the noise contained in each sound that the algorithm was able to discriminate were presented as a percentage of all noise. The relationship between age and sensitivity of wheeze recognition was analyzed using the Jonckheere–Terpstra test [29, 30]. P-value of $<0.05$ was considered statistically significant.

## Results

### Participant characteristics

Table 1 lists the participant characteristics.

**Table 1. Participant characteristics.**

| Characteristic | | Value |
|---|---|---|
| **Age, mean (SD)** | | 57.5 months (43.1 months) |
| | **2 months–11 months, n (%)** | 30 (14) |
| | **12 months–23 months, n (%)** | 26 (12) |
| | **24 months–35 months, n (%)** | 37 (17) |
| | **36 months–47 months, n (%)** | 12 (6) |
| | **48 months–59 months, n (%)** | 11 (5) |
| | **60 months–71 months, n (%)** | 30 (14) |
| | **72 months–83 months, n (%)** | 12 (6) |
| | **84 months–95 months, n (%)** | 6 (3) |
| | **96 months–107 months, n (%)** | 15 (7) |
| | **108 months–119 months, n (%)** | 9 (4) |
| | **120 months–131 months, n (%)** | 11 (5) |
| | **132 months–143 months, n (%)** | 9 (4) |
| | **144 months–155 months, n (%)** | 6 (3) |
| | **Total, n (%)** | 214 (100) |
| **Gender, (Male:Female)** | | 138:76 |
| **Height, mean (SD), range** | | 101.7 cm (25.4 cm), range: 56.7 cm to 160.5 cm |
| **Weight, mean (SD), range** | | 17.7 kg (9.3 kg), range: 5.2 kg to 46.0 kg |

## Classification of recorded sounds and wheezes characteristics

Table 2 shows the classification of recorded sounds of all lung sound samples and wheeze sound characteristics. If the wheeze contained essentially a single frequency, the wheeze was classified as monophonic wheeze. If it contained several frequencies, it was classified as polyphonic wheeze [31].

## Number of local maximum points of wheeze sounds

Table 3 shows the number of local maximum points of wheeze sounds, totaling 813 in all recorded sounds. A total of 304 wheezes were found to have one local maximum point and accounted for 37.4% of all wheeze sounds. In addition, 206 wheezes had two local maximum points and accounted for 25.3% of all wheeze sounds. Overall, less than three local maximum points accounted for >62.7% of all wheeze sounds.

**Table 2. Classification of sounds in all recorded sound files.**

| 1. Characteristics of wheeze sounds | | |
|---|---|---|
| | Frequency, mean (SD), range | 422 Hz (233 Hz), range: 100 Hz to 1380 Hz |
| | Intensity, mean (SD), range | 18.9 dB (7.1 dB), range: 3.0 dB to 44.0 dB |
| | Duration, mean (SD), range | 388 ms (245 ms), range: 100 ms to 1616 ms |
| **2. Type of wheeze sounds** | | |
| | Monophonic wheeze, n (%) | 304 (37.4) |
| | Polyphonic wheeze, n (%) | 509 (62.6) |
| **3. Noise** | | |
| | Nasal congestion, n (%) | 115 (20.6) |
| | Patient's crying or voice, n (%) | 138 (24.8) |
| | Physician's voice, n (%) | 234 (42.0) |
| | Ambient crying or voice, n (%) | 70 (12.6) |

**Table 3. Number of local maximum points of wheeze sounds in all recorded sounds.**

| Number of local maximum points | Number of wheeze sounds | Rate (%) |
|:---:|:---:|:---:|
| 1 | 304 | 37.4 |
| 2 | 206 | 25.3 |
| 3 | 107 | 13.2 |
| 4 | 88 | 10.8 |
| 5 | 52 | 6.4 |
| 6 | 20 | 2.5 |
| 7 | 11 | 1.4 |
| 8 | 7 | 0.9 |
| 9 | 5 | 0.6 |
| 10 | 6 | 0.7 |
| 11 | 4 | 0.5 |
| 12 | 3 | 0.3 |

## Image of the sound with the consecutive maximum value in step 4

Fig 3 presents a case of a consecutive maximum value detected by the algorithm on the spectrogram. Continuous local maximum values selected in step 4 included wheeze sounds and other noises (voices, ambient sounds, crying, and heartbeat sounds). The number of red squares is 315 at the consecutive large maximum point, as detected by the algorithm at 27.88 s. The white square indicates wheeze sounds. Wheeze sounds are shown in eight places in this file. Eight consecutive large maximum points were detected as wheeze sounds by the algorithm in this file and 293 points were other noises.

## Accuracy of wheeze recognition

Table 4 displays the results of wheeze recognition using the wheeze detection algorithm. The sensitivity, specificity, PPV, and NPV for wheeze recognition in all data files were 100%, 95.7%, 90.3%, and 100%, respectively.

## Influence of age on the sensitivity of wheeze detection

The sensitivity of wheeze recognition was 100% when tested using wheeze files from patients of all ages. Therefore, the sensitivity of wheeze detection was not influenced by age (p = 1).

## Automatic differentiation of wheezes from other sounds using the wheeze detection algorithm

Fig 4 shows the sound spectrograms of recorded sounds, such as monophonic wheezes, polyphonic wheeze, heartbeats, voices, and crying. The white bar indicates wheezes. Arrows

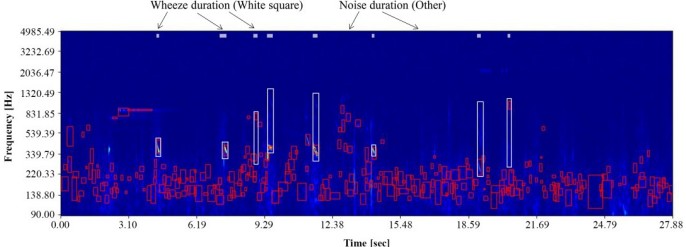

**Fig 3. Consecutive maximum values detected on a spectrogram in step 4 of a newly developed wheeze recognition algorithm for children.**

**Table 4. Results per file obtained using the newly developed wheeze recognition algorithm for children.**

|  | Specialist's diagnosis | |  |
|---|---|---|---|
|  | **Wheeze** | **Non-wheeze sounds** |  |
| **Identified as wheeze sounds** | [a]TP = 65 | [b]FP = 7 | PPV 90.3% (65/72) |
| **Identified as non-wheeze sounds** | [c]FN = 0 | [d]TN = 142 | NPV 100.0% (142/142) |
| **Results** | Sensitivity 100.0% (65/65) | Specificity 95.7% (142/149) |  |

[a]true positive

[b]false positive

[c]false negative

[d]true negative

indicate heartbeat sounds, voices, and crying. Wheezes were automatically detected, whereas heartbeat sounds, voices, and crying were automatically identified as non-wheeze sounds by the wheeze recognition algorithm.

## Discussion

Wheezes in young children, including infants, were successfully detected using the wheeze recognition algorithm. This algorithm could precisely discriminate wheezes from other noises in an environment with various sounds. Furthermore, based on the characteristics of wheezes, the automatic wheeze recognition algorithm could detect mild wheezes in crying infants recorded for 30 s in a pediatric consultation room.

Studies on the automated wheeze detection have been used in various clinical conditions [32, 33]. Development of these detection systems has been conducted in the past three decades. A meta-analysis found that computerized lung sound analysis had relatively high sensitivity and specificity in a small number of studies [33, 34]. Although wheeze detection systems have been successfully implemented, they have not been used clinically in children (including infants). The following problems have been encountered while using automated wheeze detection systems [35–37].

First, a short examination time is required to accurately detect wheezes, and a simple procedure should be clinically used in small children. In small children, including infants, recording lung sounds without crying, moving, or being distracted by the attached adhesive pads or belt is difficult. A method that recorded in 30 s period was selected by attaching a microphone to the chest wall by hand. In our study sample consisting 214 children including 30 infants, the sensitivity of the wheeze detection using our algorithm was not affected by age.

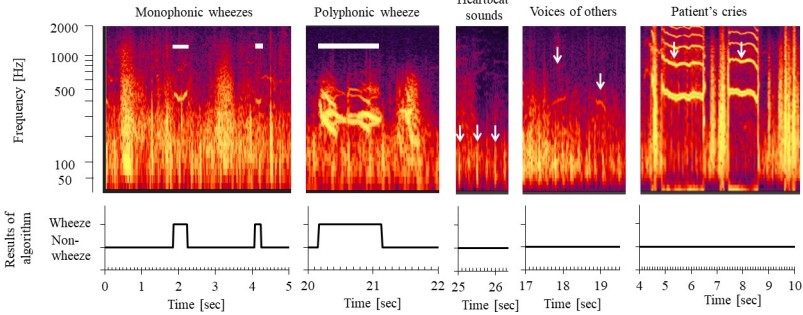

**Fig 4. Automatic discrimination of wheezes from other sounds using the newly developed wheeze detection algorithm for children.**

Second, intensity of wheezes varies among children. Wheezes are continuous adventitious lung sounds that are superimposed on breath sounds. According to new definitions in the current Computerized Respiratory Sound Analysis guidelines, the dominant frequency of a wheeze is usually > 100 Hz, with a duration of > 100 ms [31]. The most common features of detecting wheezes are the use of different wheeze peak shapes in the time–frequency plane, such as amplitude spectrum, continuity, spread, sparseness, and kurtosis. Continuous local maximum points of intensity in the spectra, which are considered as the most common features associated with wheezes during lung analysis, were analyzed. The wheeze spectra and spectrograms have many local maximum points when using FFT. Other technologies, such as PulmoTrack®, provide respiratory rates, inspiratory/expiratory time ratios, wheeze rate during the recording duration (Wz%), and duration of the wheeze. PulmoTrack® detected > 3 local maximum points. However, <3 local maximum points accounted > 62.7% of all wheeze sounds in this study, and our wheeze recognition algorithm could detect > 1 local maximum points. Moreover, this algorithm could also detect mild wheezes. Consequently, it exhibited higher sensitivity than other wheezes detection technologies. In addition, a highly precise noise canceling technology should be developed for clinical use of young children.

Third, recording lung sounds in a noisy clinic requires more rigorous post-processing than does recording in a quiet room to compensate for the noise present in the acoustic signal. Therefore, the efficiency of classification algorithms may differ. These inconsistencies would lead to difficulties in interpreting and translating study outcomes, and they have prevented the clinical use of computerized lung sound analysis devices, especially in children [31]. To improve the accuracy of the algorithm for automatically detecting wheezing, various methods have been developed with the aim of eliminating the influence of human voices and various environmental sounds, but they have not been put into practical use [38–40]. Algorithms, such as neural networks, vector quantization, or Gaussian mixture model classification systems, and support vector machines, have been used to analyze spectral features. A support vector machine is a supervised machine-learning algorithm that can be used for both classification and regression [35, 36]. The presence of wheezes can be identified using a decision tree with classifiers of other noises.

The decision tree is a method that can classify sounds according to detailed differences in sound features. Heartbeat sounds typically last for <100 ms. Voices and other sounds produced noises of higher decibel levels on the environment microphone than wheezes on the lung sound microphone. Crying was louder on the lung sound microphone than that on the environment microphone but had different continuous pattern ranges against wheeze sounds. Therefore, non-wheeze sounds could be automatically distinguished from wheezes using the wheeze detection algorithm. We discriminated wheezes from environmental noise based on the different features of wheeze sounds. Moreover, non-wheeze sounds could be automatically distinguished from other noises in a noisy pediatric consulting room.

This study has some limitations with regard to the validation of the algorithm in the clinical setting of other hospitals and clinics and at home. In addition, a real-time wheeze detection system has not been developed for clinical use. In the near future, we will collect data at multiple facilities in other countries and create and verify algorithms with higher robustness.

Wheezing often occurs in the absence of a doctor, such as during the night, at home, and/or during exercise, and possibly even in the absence of a parent. Our new home medical device, equipped with a highly accurate algorithm that is not affected by environmental noise, can easily detect wheezing and may be able to properly detect attacks at home in the absence of a doctor. Therefore, this new medical device is able to improve the safety of small children with respiratory illnesses.

## Conclusions

We successfully developed a wheeze recognition algorithm that could automatically detect wheezes for clinical use, using the lung sound analysis in children and infants. We successfully discriminated wheezes from other noises, such as heartbeats, voices, and crying, using the wheeze detection algorithm in a noisy pediatric consulting room. In the future, we hope to validate this algorithm in a larger number of patients. This practical implementation may provide beneficial information for physicians and parents of children and infants. We would like to create a novel home medical device equipped with this algorithm, which could contribute to the improvement of the safety of children with asthma.

## Acknowledgments

The authors express their immense gratitude to the children and parents who consented to participate in this study.

## Author Contributions

**Conceptualization:** Chizu Habukawa, Katsumi Murakami.

**Data curation:** Chizu Habukawa.

**Formal analysis:** Chizu Habukawa, Katsumi Murakami.

**Funding acquisition:** Chizu Habukawa, Katsumi Murakami.

**Investigation:** Chizu Habukawa.

**Methodology:** Chizu Habukawa.

**Project administration:** Chizu Habukawa.

**Resources:** Naoto Ohgami, Naoki Matsumoto, Kenji Hashino, Kei Asai, Tetsuya Sato.

**Software:** Chizu Habukawa.

**Supervision:** Katsumi Murakami.

**Validation:** Chizu Habukawa, Katsumi Murakami.

**Visualization:** Chizu Habukawa.

**Writing – original draft:** Chizu Habukawa.

**Writing – review & editing:** Chizu Habukawa, Katsumi Murakami.

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
