## [Decision Letter · Decision Letter 0]

11 Jun 2020

PONE-D-20-02208

A wheeze recognition algorithm for practical implementation in children

PLOS ONE

Dear Dr. Habukawa,

Thank you for submitting your manuscript to PLOS ONE. After careful consideration, we feel that it has merit but does not fully meet PLOS ONE’s publication criteria as it currently stands. Therefore, we invite you to submit a revised version of the manuscript that addresses the points raised during the review process.

We have managed to finalize the reviewing process. I am confident that if you will follow the provided comments, the manuscript will reach a publishable form. In addition, I recommend you to pay attention to the writing style, which might be improved (the support of a native English speaker would be a plus). Furthermore, equations from L183-186 could be rewritten in a more mathematical way and general statements will benefit if adequate references are mentioned (e.g. L188 a reference for the Jonckheere–Terpstra test). More examples of spectrograms organized by levels of severity will increase the scientific credibility of the research.

We look forward to receiving your revised manuscript.

Kind regards,

Daniel Dunea, Ph.D.

Academic Editor

PLOS ONE

Journal Requirements:

'I have read the journal's policy and the authors of this manuscript have the following competing interests: Chizu Habukawa and Katsumi Murakami received a research grant from Omron Health Care Corporation. Naoto Ohgami, Naoki Matsumoto, Kenji Hashino, Kei Asai, and Tetsuya Sato are employees of Omron Healthcare Co., Ltd.'

We note that one or more of the authors are employed by a commercial company: Omron Healthcare Co. Ltd

Reviewers' comments:

Reviewer's Responses to Questions

**Comments to the Author**

1. Is the manuscript technically sound, and do the data support the conclusions?

Reviewer #1: Yes

Reviewer #2: Yes

Reviewer #3: Yes

2. Has the statistical analysis been performed appropriately and rigorously? 

Reviewer #1: Yes

Reviewer #2: Yes

Reviewer #3: Yes

3. Have the authors made all data underlying the findings in their manuscript fully available?

Reviewer #1: Yes

Reviewer #2: Yes

Reviewer #3: Yes

4. Is the manuscript presented in an intelligible fashion and written in standard English?

Reviewer #1: No

Reviewer #2: Yes

Reviewer #3: Yes

5. Review Comments to the Author

Reviewer #1: The authors describe an automatic algorithm for the detection of wheezing from recorded breath sounds. Although the study is conceptually highly technical, the authors have done an excellent job of explaining their methods and results.

However, their presentation of the methods and results requires a lot of improvement. Several sections of the manuscript also need to be clarified and/or written better.

Abstract: the sentence beginning towards the end of line 28 and ending in line 31 can be simplified to: "Files containing recorded lung sounds were assessed by two specialist physicians and divided into two groups: 65 were designates as 'wheeze' files and 149 designated 'no-wheeze' files". It is confusing as currently written.

Introduction: in line 52, you mean 'highly subjective'; in line 74 you mean 'which can automatically recognise wheezes.

Methods: the whole methods section needs to be reorganised; it is currently cluttered and includes information which belongs elsewhere. See details below.

- line 85, replace 'data' with 'date'.

- most of what you have presented under 'Participants' are actually results and should be moved to the results section and summarised in a new Table 1 which should be added before the current one. Please look at how a Table 1 has been presented in other published manuscripts.

- line 96: briefly describe what the guidelines say.

- line 96: the second half of that line does not make any sense or fit with the rest of the paragraph.

- 'Study design' section does not describe the study design, but instead describes the study procedures

- line 112: what does 'accorded' mean in this context; please rewrite this sentence for clarity

- line 112: second part of this line (which ends on line 113) does not make sense either; please rewrite for clarity

- line 130: the definition of a wheeze seems quite broad; for example, a sound of 1024Hz and 512ms (or even 1024MHz and 512s) would meet the criteria specified, i.e. >100Hz and >100ms in duration), but would almost certainly not be a wheeze. Please clarify whether there are any upper limits on those ranges.

- lines 155 to 157 are the most technical part of the description of this study but no clear explanation of what they mean is provided

- lines 171 and 172 - you mean 'identified' instead of 'recognized'.

- line 188 - provide some explanation of what the Jonckheere-Terpstra test is and what it does, including a citation

Results

- Table 1 is incomplete; you need to include the measures of dispersion (e.g. SDs) for the averages presented. You also need to indicate whether the characteristics indicated in (2) and (3) are mutually exclusive, and what proportions of sounds had those features.

Reviewer #2: The current manuscript is a test of the validity of an automatic wheeze detection algorithm in children. Because wheeze is an indicator of worsening respiratory diseases and respiratory diseases are extremely common in children, there is a need to have an efficient and accurate system with potential as a screening tool or a measure of response to treatment. Although different applications and potential implementation of this algorithm are still to be assessed, the current manuscript provides and outstanding proof of concept that is well presented. Rather than a discussion of problems, issues, or major revisions; the remainder of the review concerns some areas for clarification.

There is clear value in providing a method that discriminates wheeze sounds from other background noise. Is there evidence that clinician errors in wheeze detection are affected by ambient noise. Also, parents or family members reporting wheeze is an unreliable clinical resource, are there significant efforts to train parents in wheeze detection and differentiation. The evidence presented for the tested algorithm is strong, well designed, and effectively analyzed. However, how does this system improve upon low tech clinical approaches to wheeze detection. Accuracy, efficiency, rapidity, reliability, and convenience are all possible reasons why this approach has value. A bit more detail on the basic rationale would be helpful.

As a very minor detail. On p 9, the phrase "these feature values were combined, evaluated, and classified." A bit more detail defining exactly what and how these three components were achieved.

What was the correspondence between the two specialist physicians? A basic inter-rater reliability metric would be helpful.

I am more of a measurement person and not a clinical expert, so this may be an ignorant question: but is there any clinical implication of a monophonic versus polyphonic wheeze?

I appreciate the need to create a algorithm that is robust to ambient sounds. I would like some specifics as the the next steps for this work. Home and clinic testing is reasonable. What would the next steps be?

I found figures three and four to be extremely helpful. Congratulations on using figures to communicate clearly.

Reviewer #3: Material and method: it is good to add the country name of the hospital where the study was conducted and a short description of the hospital - is it a tertiary level ? how many children visit per day or per month?

Study design: need major revision of this section. who listened and classified recorded sound file? their qualification, training and standardisation process? how did solve the discordant sound files classification? Ref 18 need to check again.

6. PLOS authors have the option to publish the peer review history of their article (what does this mean?). If published, this will include your full peer review and any attached files.

Reviewer #1: No

Reviewer #2: No

Reviewer #3: Yes: Salahuddin Ahmed

---

## [Author Response · Author response to Decision Letter 0]

30 Jul 2020

Dear Dr. Daniel Dunea, Ph.D.

Academic Editor

PLOS ONE

Dear Dr. Daniel Dunea

Thank you for sending the helpful comments from the reviewers on our manuscript. We have addressed all of the comments and revised the manuscript as described below. We hope that you will find these revisions acceptable.

Comments to the Author

I recommend you to pay attention to the writing style, which might be improved (the support of a native English speaker would be a plus). 

Furthermore, equations from L183-186 could be rewritten in a more mathematical way and general statements will benefit if adequate references are mentioned (e.g. L188 a reference for the Jonckheere–Terpstra test). 

More examples of spectrograms organized by levels of severity will increase the scientific credibility of the research.

Response to Dr. Daniel Dunea

Thank you for your helpful comments.

We have adapted the article to the writing style of the journal and have received English language support and proofreading (confirmation attached).

We have revised this section and added references as follows:

The results fell into one of the following four categories: actual positives that were correctly predicted as positives (true positives, TP); actual positives that were wrongly predicted as negatives (false negatives, FN); actual negatives that were correctly predicted as negatives (true negatives, TN); and actual negatives that were wrongly predicted as positives (false positives, FP). We analyzed the sensitivity, specificity, positive predictive value (PPV), and negative predictive value (NPV) using the results of the wheeze recognition algorithm in all data files [26,27].

(L178–184)

Sensitivity is the probability that the algorithm correctly recognized as wheeze for patients who diagnosed as wheeze by specialist. Specificity is the probability that the algorithm correctly recognized as non-wheeze for patients who diagnosed as non-wheeze by specialist. 

 PPV is the probability that files that were identified as “wheeze” files by the specialists were also recognized as “wheeze” files by the algorithm. PNV is the probability that files that were identified as “non-wheeze” files by the specialists were also recognized as “non-wheeze” files by the algorithm. (L184–192)

Sensitivity =(number of True positive)/(number of Condition Positive)=(TP )/(TP +FN )

Specificity =(number of True Negative)/(number of Condition Negative)=(TN )/(TN+FP )

PPV=(number of True Positive)/(number of Predicted condition Positive)=(TP )/(TP +FP )

 NPV=(number of True Negative)/(number of Predicted condition Negative)=(TN )/(TN+FN )

Statistical analysis was performed using R software, version 3.4.1 [28] (L193). The relationship between age and sensitivity of wheeze recognition was analyzed using the Jonckheere–Terpstra test [29,30] (L197–199).

[28] R Core Team. R: A language and environment for statistical computing. 2018; Vienna, Austria, R Foundation for Statistical Computing. Available from: https://www.R-project.org/

[29] Julia Wisniewski, Rachana Agrawal, et al. Sensitization to Food and Inhalant Allergens in Relation to Age and Wheeze Among Children with Atopic Dermatitis , Clin Exp Allergy, 2013; 43(10): 1160–1170

[30] Hiroshi Hashizume, Shin ichi Konno, et al. Japanese orthopaedic association back pain evaluation questionnaire (JOABPEQ) as an outcome measure for patients with low back pain: reference values in healthy volunteers. J Orthop Sci, 2015; 20: 264–280

In addition, in Figure 4, we have shown the severe wheeze sounds as polyphonic wheeze and the mild wheeze sounds as monophonic wheeze. (L254-255)

 

Review 1 Comments to the Author

The authors describe an automatic algorithm for the detection of wheezing from recorded breath sounds. Although the study is conceptually highly technical, the authors have done an excellent job of explaining their methods and results.

However, their presentation of the methods and results requires a lot of improvement. Several sections of the manuscript also need to be clarified and/or written better.

Response to Referee 1

Thank you for sending the helpful comments from the reviewers on our manuscript. 

We have addressed all of the comments and have revised the manuscript as described below. We hope that you will find these revisions acceptable. We have adapted the article according to the writing style of the journal and have received English language support and proofreading (confirmation attached). 

Abstract: the sentence beginning towards the end of line 28 and ending in line 31 can be simplified to: "Files containing recorded lung sounds were assessed by two specialist physicians and divided into two groups: 65 were designates as 'wheeze' files and 149 designated 'no-wheeze' files". It is confusing as currently written.

Reply 1

We have corrected these sentences (L28–30). 

Files containing recorded lung sounds were assessed by two specialist physicians and divided into two groups: 65 were designated as “wheeze” files, and 149 were designated as “no-wheeze” files.

Introduction: in line 52, you mean 'highly subjective'; in line 74 you mean 'which can automatically recognise wheezes.

Reply 2

We have corrected these sentences. 

Results of auscultation are highly subjective and cannot be easily reproduced because of differences in doctors’ experience and perceptual abilities, which cause differences in assessment. (L52)

Few reported algorithms were available for clinical use in a noisy environment, such as a pediatric consultation room or at home, which can automatically recognize wheezes. (L73)

Methods: the whole methods section needs to be reorganised; 

it is currently cluttered and includes information which belongs elsewhere. See details below.

- line 85, replace 'data' with 'date'.

Reply 3

We have corrected these sentences (L87).

The study protocol was approved by the ethics committee of the hospital [approval number 2016-22(2); approval date: January 19, 2016].

- most of what you have presented under 'Participants' are actually results and should be moved to the results section and summarised in a new Table 1 which should be added before the current one. Please look at how a Table 1 has been presented in other published manuscripts.

Reply 4

We have rewritten the Participants section and have added a new table (Table 1) to the Results section. 

Table 1. Participant characteristics.

Characteristic Value

Age, mean (min–max) 49 months (2 months–155 months)

 2 months–11 months, n 30

 12 months–23 months, n 26

 24 months–35 months, n 37

 36 months–47 months, n 12

 48 months–59 months, n 11

 60 months–71 months, n 30

 72 months–83 months, n 12

 84 months–95 months, n 6

 96 months–107 months, n 15

 108 months–119 months, n 9

 120 months–131 months, n 11

 132 months–143 months, n 9

 144 months–155 months, n 6

 Total, n 214

Male sex, n (%) 138 (64)

Height, mean (min–max) 91.3 cm (52.6 cm–131.5 cm)

Weight, mean (min–max) 13.9 kg (4 kg–30 kg)

- line 96: briefly describe what the guidelines say.

Reply 5

We have added an explanation of the treatment guidelines (L89–95).

 All participants aged ≥6 years were diagnosed with bronchial asthma, and their asthmatic severities were classified as mild asthma according to the Japanese Pediatric Guideline for the Treatment and Management of Asthma 2017 [17]. These participants were treated with leukotriene receptor antagonist and/or inhaled corticosteroid in accordance with the guidelines [17] Participants aged ≤5 years had reported at least two episodes of wheezes and had been treated with leukotriene receptor antagonist or without medicine for long-term management.

- line 96: the second half of that line does not make any sense or fit with the rest of the paragraph.

Reply 6

We have added an explanation of this sentence. (L93–95)

 Participants aged ≤5 years had reported at least two episodes of wheezes and had been treated with leukotriene receptor antagonist or without medicine for long-term management.

- 'Study design' section does not describe the study design, but instead describes the study procedures

Reply 7

We have corrected it “Study procedures” (L100).

- line 112: what does 'accorded' mean in this context; please rewrite this sentence for clarity

- line 112: second part of this line (which ends on line 113) does not make sense either; please rewrite for clarity

Reply 8

We have rewritten these sentences (L104–107).

 The recorded lung sounds (with or without wheezes) were then listened to independently by two specialist physicians and then confirmed and classified in accordance with the previous methods. Judgments were agreed between two physicians [18].

- line 130: the definition of a wheeze seems quite broad; for example, a sound of 1024Hz and 512ms (or even 1024MHz and 512s) would meet the criteria specified, i.e. >100Hz and >100ms in duration), but would almost certainly not be a wheeze. Please clarify whether there are any upper limits on those ranges.

Reply 9

We have rewritten the sentences and added references. (L124–128)

Following the Computerized Respiratory Sound Analysis guidelines, the dominant frequency of a wheeze is >100 Hz and the duration is >100 ms [19]. Furthermore, a previous report described the frequency range of a typical wheeze as between 100 and 5000 Hz [20]. The maximum duration of a wheeze is within the expiratory duration.

[20] N. Meslier, G. Charbonneau, J-L. Racineux. Wheezes. Eur Respir J. 1995; 8: 1942-1948

- lines 155 to 157 are the most technical part of the description of this study but no clear explanation of what they mean is provided

Reply 10

We have rewritten these sentences and added references (L152–155).

Fast Fourier transform (FFT), the most popular acoustic analysis method, was used. FFT analyzes the intensity for each frequency of sound data. The sound data were preprocessed in a Hamming window of 4096 points (372 ms), and processing was repeated each time the sound data increased by 128 points (18 ms). [23–25]

[23] Styliani A. Taplidou, Leontios J. Hadjileontiadis. Wheeze detection based on time-frequency analysis of breath sounds. Computers in Biology and Medicine. 2007; 37, 1073-1083 

[24] Tatsuya Yoshimasu, Mitsumasa Kawago, Yoshitaka Okamura. et al. Fast Fourier Transform Analysis of Pulmonary Nodules on Computed Tomography Images from Patients with Lung Cancer. Ann Thorac Cardiovasc Surg. 2015; 21, 1-7.

[25] L. Bentur, R. Beck, M. Shinawi, T. Naveh, N. Gavriely. Wheeze monitoring in children for assessment of nocturnal asthma and response to therapy. Eur Respir J. 2003; 21, 621-626

- lines 171 and 172 - you mean 'identified' instead of 'recognized'.

Reply 11

We have corrected this error (L171–172, L257 and Table 4).

Finally, if at least one wheeze sound was heard in a file, the file was identified as a wheeze file, whereas if no wheeze sound was heard in a file, it was identified as a no-wheeze file. (L171–172)

- line 188 - provide some explanation of what the Jonckheere-Terpstra test is and what it does, including a citation

Reply 12

We have added references [29,30] (L198).

The Jonckheere–Terpstra test is a non-parametric test used to detect trends in increases and decreases for time series data. 

[29] Julia Wisniewski, Rachana Agrawal, et al. Sensitization to Food and Inhalant Allergens in Relation to Age and Wheeze Among Children with Atopic Dermatitis , Clin Exp Allergy, 2013; 43(10): 1160–1170

[30] Hiroshi Hashizume, Shin ichi Konno, et al. Japanese orthopaedic association back pain evaluation questionnaire (JOABPEQ) as an outcome measure for patients with low back pain: reference values in healthy volunteers. J Orthop Sci, 2015; 20: 264–280N. Meslier, G. Charbonneau, J-L. Racineux. Wheezes. Eur Respir J. 1995; 8: 1942-1948

Results

- Table 1 is incomplete; you need to include the measures of dispersion (e.g. SDs) for the averages presented. You also need to indicate whether the characteristics indicated in (2) and (3) are mutually exclusive, and what proportions of sounds had those features.

Reply 13

We have corrected this table (Table 2 in the revised manuscript). The table now describes the characteristics of wheeze sounds, including the mean ± SD and range, and the ratio of the noise contained in each sound that the algorithm was able to discriminate. (L213, Results) 

Table 2. Classification of sounds in all recorded sounds.

1. Characteristics of wheeze sounds

 Frequency, mean ± SD (min–max) 422 Hz ± 233 Hz (100 Hz–1380 Hz)

 Intensity, mean ± SD (min–max) 18.9 dB ± 7.1 dB (3.0 dB–44.0 dB)

 Duration, mean ± SD (min–max) 388 ms ± 245 ms (100 ms–1616 ms)

2. Type of wheeze sounds

 Monophonic wheeze, n (%) 304 (37.4)

 Polyphonic wheeze, n (%) 509 (62.6)

3. Noise

 Nasal congestion, n (%) 115 (20.6)

 Patient’s crying or voice, n (%) 138 (24.8)

 Physician’s voice, n (%) 234 (42.0)

 Ambient crying or voice, n (%) 70 (12.6)

 

Reviewer #2: 

The current manuscript is a test of the validity of an automatic wheeze detection algorithm in children. Because wheeze is an indicator of worsening respiratory diseases and respiratory diseases are extremely common in children, there is a need to have an efficient and accurate system with potential as a screening tool or a measure of response to treatment. 

Although different applications and potential implementation of this algorithm are still to be assessed, the current manuscript provides and outstanding proof of concept that is well presented. Rather than a discussion of problems, issues, or major revisions; the remainder of the review concerns some areas for clarification.

Response to Referee 2

Thank you for your helpful comments on our manuscript. We have addressed all of the comments and revised the manuscript as described below. We hope that you will find these revisions acceptable.

There is clear value in providing a method that discriminates wheeze sounds from other background noise. Is there evidence that clinician errors in wheeze detection are affected by ambient noise. 

Reply 14

This concept has been accepted that clinicians are able to listen to wheezing when they auscultate using a stethoscope, even in a noisy examination room, because they are able to automatically cancel environmental noise sounds. Humans are able to eliminate background noise and focus on specific sounds; however, machines do not have that capability, therefore, algorithms for noise reduction are required for clinical use. 

We have added the following sentences and references (L306-309):

To improve the accuracy of the algorithm to automatically detect wheezing, various methods have been employed to eliminate the influence of human voices and various other environmental sounds, but they have not been put into practical use. [38–40].

[38] Shih-Hong Li, Bor-Shing Lin, Bor-Shyh Lin, et al. Design of Wearable Breathing Sound Monitoring System for Real-Time Wheeze Detection. Sensors. 2017; 17, 171.

[39] Andrès E, Gass R, Hentzler A, et al. Respiratory sound analysis in the era of evidence-based medicine and the world of medicine 2.0. Journal of Medicine and Life. 2018; 11, issue2, 89-106

[40] Lia C Puder, Silke Wilitzki, Christoph Bührer, Hendrik S Fischer1, Gerd Schmalisch. Computerized wheeze detection in young infants: comparison of signals from tracheal and chest wall sensors. Physiol. Meas. 2016; 37, 2170-2180 

- Also, parents or family members reporting wheezing is an unreliable clinical resource efforts to train parents in wheeze detection and differentiation. The evidence presented for the tested algorithm is strong, well designed, and effectively analyzed. However, how does this system improve upon low tech clinical approaches to wheeze detection. Accuracy, efficiency, rapidity, reliability, and convenience are all possible reasons why this approach has value. A bit more detail on the basic rationale would be helpful.

Reply 15

We have added the following to the Discussion section (L330–335).

Wheezing often occurs in the absence of a doctor, such as during the night, at home and/or during exercise, and possibly even in the absence of a parent. Our new home medical device, equipped with a highly accurate algorithm that is not affected by environmental noise can easily detect wheezing and may be able to properly detect attacks at home in the absence of a doctor. Therefore, this new medical device is able to improve the safety of small children with respiratory illnesses. 

- As a very minor detail. On p 9, the phrase "these feature values were combined, evaluated, and classified." A bit more detail defining exactly what and how these three components were achieved.

Reply 16

We have rewritten the sentence as follows (L168-170).

The feature values of the wheezing candidates selected in step 4 were calculated, and the classification model was created using a machine-learning method (decision tree).

What was the correspondence between the two specialist physicians? 

A basic inter-rater reliability metric would be helpful.

Reply17 

We have added the following sentence and reference (L107).

 Judgments were agreed between two physicians [18].

[18] Puder LC, Fischer HS, Wilitzki S, Usemann J, Godfrey S, Schmalisch G. BMC Validation of computerized wheeze detection in young infants during the first months of life. Pediatr. 2014 9;14: 257.

I am more of a measurement person and not a clinical expert, so this may be an ignorant question: but is there any clinical implication of a monophonic versus polyphonic wheeze?

Reply 18

Wheezing is considered monophonic when only one pitch is heard and is considered polyphonic when multiple frequencies are perceived simultaneously. Polyphonic wheezing involves more severe bronchial constriction than monophonic wheezing.

References: 

[3] Forgacs P. The functional basis of pulmonary sounds. Chest 1978; 73: 399-405.

Shim CS, Williams MH. Relationship of wheezing to the severity of obstruction in asthma. Arch Intern Med 1983; 143: 890-2.

Baughman RP, Loudon RG. Quantitation of wheezing in acute asthma. Chest 1984; 86: 

718-22.

I appreciate the need to create a algorithm that is robust to ambient sounds. I would like some specifics as the next steps for this work. Home and clinic testing is reasonable. What would the next steps be?

Reply 19

We have added the following sentences to the Discussion section (L327-329).

In the near future, we will collect data at multiple facilities in other countries and create and verify algorithms with higher robustness.

We would like to create a novel home medical device equipped with this algorithm, which could contribute to the improvement of the safety of children with asthma. 

I found figures three and four to be extremely helpful. Congratulations on using figures to communicate clearly.

Reply 20

We really appreciate your comments.

 

Reviewer #3: 

Response to Referee 3

Thank you for your helpful comments on our manuscript. We have addressed all of the comments and revised the manuscript as described below. We hope that you will find these revisions acceptable.

Material and method: it is good to add the country name of the hospital where the study was conducted and a short description of the hospital - is it a tertiary level ? 

Reply 21

We have corrected this sentence (L81–84).

All participants were outpatients who attended the Minami Wakayama Medical Center, the secondary emergency general hospital of the National Hospital Organization, located in Wakayama, Japan between September 27, 2016, and October 25, 2017.

how many children visit per day or per month?

Reply 22

On average, 70 patients with allergic disease visit physicians every day at the Minami Wakayama Medical Center. In this study, we recorded wheezing during a 13-month period. 

We have added the dates in the Participants section (L81–84). 

All participants were outpatients who attended the Minami Wakayama Medical Center, the secondary emergency general hospital of the National Hospital Organization, located in Wakayama, Japan between September 27, 2016, and October 25, 2017.

Study design: need major revision of this section. who listened and classified recorded sound file? 

Reply 23

We have corrected the sentences in the Participants and the Study procedures sections and have added a new table (Table 1).

The recorded lung sounds (with or without wheezes) were then listened to independently by two specialist physicians and then confirmed and classified in accordance with the previous methods. Judgments were agreed between two physicians [18]. (L104–107).

their qualification, training and standardisation process? 

how did solve the discordant sound files classification? 

Reply 24

Two of the authors are specialist physicians in pediatrics and allergology, who have been working for >30 years. In this study, wheeze judgments were agreed between two physicians.

Ref 18 need to check again.

Reply 25

We have corrected reference 18:

[18] Puder LC, Fischer HS, Wilitzki S, Usemann J, Godfrey S, Schmalisch G. BMC Validation of computerized wheeze detection in young infants during the first months of life. Pediatr. 2014 9;14:257.

---

## [Decision Letter · Decision Letter 1]

7 Sep 2020

PONE-D-20-02208R1

A wheeze recognition algorithm for practical implementation in children

PLOS ONE

Dear Dr. Habukawa,

Thank you for submitting your manuscript to PLOS ONE. After careful consideration, we feel that it has merit but does not fully meet PLOS ONE’s publication criteria as it currently stands. Therefore, we invite you to submit a revised version of the manuscript that addresses the points raised during the review process.

We have received the reviewers' comments. Based on their evaluation and my opinion after reading the revised manuscript, I recommend minor revision. Please address the comments raised by one of the reviewers and perform a quality control on the revised manuscript (potential typos, unclear statements, etc.) before resubmission. The scientific merit is more visible in the revised version, but few adjustments are required before publishing. 

We look forward to receiving your revised manuscript.

Kind regards,

Daniel Dunea, Ph.D.

Academic Editor

PLOS ONE

Reviewers' comments:

Reviewer's Responses to Questions

**Comments to the Author**

1. If the authors have adequately addressed your comments raised in a previous round of review and you feel that this manuscript is now acceptable for publication, you may indicate that here to bypass the “Comments to the Author” section, enter your conflict of interest statement in the “Confidential to Editor” section, and submit your "Accept" recommendation.

Reviewer #1: (No Response)

Reviewer #3: All comments have been addressed

2. Is the manuscript technically sound, and do the data support the conclusions?

Reviewer #1: Yes

Reviewer #3: Yes

3. Has the statistical analysis been performed appropriately and rigorously? 

Reviewer #1: Yes

Reviewer #3: I Don't Know

4. Have the authors made all data underlying the findings in their manuscript fully available?

Reviewer #1: Yes

Reviewer #3: Yes

5. Is the manuscript presented in an intelligible fashion and written in standard English?

Reviewer #1: Yes

Reviewer #3: Yes

6. Review Comments to the Author

Reviewer #1: The authors have responded appropriately to previous comments and the manuscript is much improved. There are a few additional changes that could improve the manuscript even further.

In Table 1, please present age, mean (SD) overall, not for different age groups, as you have done with sex, n (%). Also present mean (SD) for height and weight.

In Table 2, please remove the ± designation. You could present the characteristics as mean (SD) with the range below them e.g. 422 Hz (233 Hz) and then in the line below, range 100 Hz to 1380 Hz.

Reviewer #3: Authors addressed all comments clearly. The revised version is well written and addressed all reviewers comments. Still it is not clear to me if two physicians lung sound interpretation was not match then how they solve it.

7. PLOS authors have the option to publish the peer review history of their article (what does this mean?). If published, this will include your full peer review and any attached files.

Reviewer #1: No

Reviewer #3: **Yes: **Salahuddin Ahmed

---

## [Author Response · Author response to Decision Letter 1]

17 Sep 2020

Dear Dr. Daniel Dunea, Ph.D.

Academic Editor

PLOS ONE

Comments to the Author

We have received the reviewers' comments. Based on their evaluation and my opinion after reading the revised manuscript, I recommend minor revision. Please address the comments raised by one of the reviewers and perform a quality control on the revised manuscript (potential typos, unclear statements, etc.) before resubmission. The scientific merit is more visible in the revised version, but few adjustments are required before publishing. 

Dear Dr. Daniel Dunea

Thank you for sending the helpful comments from the reviewers on our manuscript. We have addressed all the comments and revised the manuscript as described below. We hope that you will find these revisions acceptable.

Review 1 Comments to the Author

The authors have responded appropriately to previous comments and the manuscript is much improved. There are a few additional changes that could improve the manuscript even further.

Response to Referee 1

Thank you for sending the helpful comments from the reviewers on our manuscript. 

We have addressed all the comments and have revised the manuscript as described below. We hope that you will find these revisions acceptable.

In Table 1, please present age, mean (SD) overall, not for different age groups, as you have done with sex, n (%). Also present mean (SD) for height and weight.

Reply 1

We have added SD of age, height, and weight to Table 1. In addition, we have deleted the age range because we have already mentioned this in the text pertaining to different age groups. Furthermore, we have rewritten the range for height and weight as well as gender and added the percentage for different age groups. We found a few typographical errors in Table 1, which have now been corrected. This revision has not affected the results or conclusions.

In Table 2, please remove the ± designation. You could present the characteristics as mean (SD) with the range below them e.g. 422 Hz (233 Hz) and then in the line below, range 100 Hz to 1380 Hz.

Reply 2

We have rewritten SD and range in Table 2. 

Reviewer #3: 

Response to Referee 3

Thank you for your helpful comments on our manuscript. We have addressed all the comments and revised the manuscript as described below. We hope that you will find these revisions acceptable.

Authors addressed all comments clearly. The revised version is well written and addressed all reviewers comments. Still it is not clear to me if two physicians lung sound interpretation was not match then how they solve it.

Reply 3

We have accordingly corrected this sentence in the revised manuscript (L107–108).

If the judgments of the two physicians did not match, the supervisor made the judgment [18].

---

## [Editor Report · Decision Letter 2]

18 Sep 2020

A wheeze recognition algorithm for practical implementation in children

PONE-D-20-02208R2

Dear Dr. Habukawa,

We’re pleased to inform you that your manuscript has been judged scientifically suitable for publication and will be formally accepted for publication once it meets all outstanding technical requirements.

Despite a long editorial process, the manuscript has reached the quality to be considered for publication.

For a better clarification, I would suggest that in the Abstract at Conclusions to add "...in the practical implementation of respiratory illness management at home using properly developed devices."

I wish you success in your activity!

Kind regards,

Daniel Dunea, Ph.D.

Academic Editor

PLOS ONE

---

## [Editor Report · Acceptance letter]

25 Sep 2020

PONE-D-20-02208R2 

A wheeze recognition algorithm for practical implementation in children 

Dear Dr. Habukawa:

I'm pleased to inform you that your manuscript has been deemed suitable for publication in PLOS ONE. Congratulations! Your manuscript is now with our production department. 

Kind regards, 

on behalf of

Dr. Daniel Dunea 

Academic Editor

PLOS ONE